# Copper-Free Halodediazoniation of Arenediazonium Tetrafluoroborates in Deep Eutectic Solvents-like Mixtures

**DOI:** 10.3390/molecules27061909

**Published:** 2022-03-15

**Authors:** Giovanni Ghigo, Matteo Bonomo, Achille Antenucci, Chiara Reviglio, Stefano Dughera

**Affiliations:** 1Department of Chemistry, University of Turin, Via Pietro Giuria 7, 10125 Turin, Italy; achille.antenucci@unito.it (A.A.); chiara.reviglio@edu.unito.it (C.R.); 2NIS Interdepartmental Centre and INSTM Reference Centre, Universiy of Turin, Via Gioacchino Quarello 15/a, 10125 Turin, Italy

**Keywords:** Sandmeyer reactions, deep eutectic solvents, reaction mechanism

## Abstract

Deep Eutectic Solvent (DES)-like mixtures, based on glycerol and different halide organic and inorganic salts, are successfully exploited as new media in copper-free halodediazoniation of arenediazonium salts. The reactions are carried out in absence of metal-based catalysts, at room temperature and in a short time. Pure target products are obtained without the need for chromatographic separation. The solvents are fully characterized, and a computational study is presented aiming to understand the reaction mechanism.

## 1. Introduction

In recent years, the interest in Deep Eutectic Solvents (DESs) has been growing since they are a potentially environmentally benign and sustainable alternative to conventional organic solvents [1,2,3]. Despite the increasing use of DESs in organic synthesis [4,5,6], they have scarcely been utilized in the reactions of diazonium salts. In fact, the literature shows only three very recent examples, where DESs are employed as innocent solvents [7,8,9].

In our previous paper [10] we studied the behaviour of arenediazonium tetrafluoroborates in a new DES formed by KF as a hydrogen bond acceptor (HBA) and glycerol as a hydrogen bond donor (HBD) as reaction media. A controlled decomposition of diazonium salts took place, with the formation of arenes as hydrodediazoniation products. We proposed a plausible mechanism where a relatively fast (strictly depending on the electronic effects of the substituents bound to the aromatic ring) reduction reaction occurs initiated by the formation of a glycerolate-like species. Interestingly, in some preliminary tests we found that the behaviour of 4-nitrobenzenediazonium tetrafluoroborate (**1a**) in a mixture formed by glycerol (HBD) and KBr or KCl (HBA) was completely different. In fact, a certain amount of halogenation product was obtained, still alongside the reduction product. Therefore, in the present paper, we decided to further investigate the reactivity of these mixtures. 

The transformations of arenediazonium salts into several functional groups, such as halogen, hydroxyl, and cyano, are known as Sandmeyer or Sandmeyer-type reactions (Figure 1) and have been widely used both in the fields of research and in industrial production [11,12,13,14]. 

In particular, halodediazoniation [15,16,17,18] represents an important organic transformation that converts arylamines to aryl halides via a diazonium salt intermediate and in the presence of Cu(I) as a catalyst. It must be stressed that aryl halides have always been drawing notable attention as beneficial reagents for arylation or for further aromatic core modification [19,20,21]. On these grounds, they have found a wide application in the synthesis of compound libraries, first of all because of smoothly running arylation reactions, as well as metal-catalyzed couplings. Moreover, they are an important structural motif in molecules, usually of marine origin, that exhibit interesting biological and pharmacological activity [22,23].

In recent years, this “old” reaction (proposed by Sandmeyer at the end of the 19th century [14]) has been revisited and has become an interesting research topic (Figure 1) [24,25]. In light of these, a conversion of aromatic amines via arenediazonium salts in aromatic compounds bearing a C-CF_3_ [26,27], C-B [28,29], C-Sn [30], C-S [31,32,33,34], and C-P [35,36] bonds have been proposed. Furthermore, some research groups, aiming at increasing the sustainability of the reaction, have developed interesting synthetic procedures that allow aryl halide to be obtained without the use of copper as a catalyst [37,38,39,40,41].

On this basis, in this paper we report a thorough experimental and theoretical study of halodediazoniation reactions carried out in various types of DES-like systems and in total absence of a Cu catalyst. 

## 2. Results and Discussion

As mentioned above, in one of our recent papers [10], we proposed a new DES based on glycerol and KF as an effective solvent for the green reduction of arenediazonium salts with almost quantitative yields. In that context, glycerol/KF 6:1 was unambiguously categorized as a DES system due to an unexpectedly high conductivity (31 S m^2^ mol^−1^), much higher than the values predicted by the Walden plot. Similar behaviour was found when K^+^ was replaced by Cs^+^, indicating the main role of the anion in the formation of a DES. Indeed, when F^−^ was replaced with Cl^−^ or Br^−^, the resulting systems presented, from an electrochemical point of view, a conventional behaviour. Additionally, they seemed to be ineffective in the reduction reaction whose mechanism is based on the formation of a glycerolate-like species (i.e., a glycerol molecule involved in a hydrogen bond network). This evidence pushed us to preliminarily classify glycerol/KBr 6:1 and glycerol/KCl 6:1 as “simple” salts in solvent mixtures. Similar findings were evidenced when the cation was tetrabutylammonium (TBA), characterized by a slight decrease in the conductivity of the system ascribable to the increase in the viscosity due to higher steric hindrance of TBA^+^ with respect to K^+^. Indeed, in the case of TBABr-Glycerol systems, Andruch et al. [42] reported on the formulation of three mixtures in which the molar ratio of TBABr decreased from 0.5 to 0.20. The latter, having a TBABr:Glycerol ratio of 1:4, presented a DES-like behaviour with a homogeneous aspect and a relatively low melting point (233 K). However, our attempt in the formulation of the same mixture led to unstable solutions (some salt is still present as suspension), proving the needs of a higher amount of HBD (i.e., glycerol).

In this research, we propose a more detailed analysis, also taking into account the thermal behaviour of the proposed mixtures. Conventionally, Differential Scan Calorimetry (DSC) is the technique of election to monitor the freezing/melting process, which is clearly evidenced by a positive/negative peak due to the exothermic/endothermic nature of the transition. Unfortunately, in the case of glycerol, these peaks are not detectable due to the almost heat-neutrality of the freezing process [43], and the same applies to its mixture with the investigated salts (see Appendix A). These findings prevented us from reliably employing DSC to determine the freezing point of our mixtures. Indeed, the glycerol freezing point is tabulated, but, on the other hand, no values are available for its mixture with unconventional salts. As a matter of fact, the freezing temperature of a (glycerol-based) mixture can be theoretically calculated by means of the freezing point depletion rule (ΔT = K_cr_ × *i* × m, where K_cr_ is the cryoscopic constant, 3.5 K Kg mol^−1^ for glycerol [44], *i* is maximum number of ions in which the solute could be divided, and m is the molality of the solution). Having this in mind, an ideal mixture should present a ΔT of 13 K for all systems being independent on the nature of the solute. This should give a freezing point of 278 K (T_f_ = 291 K for pure glycerol) for the mixture that is investigated here. Once ruling out the possibility of effectively employed DSC to detect the freezing point of the mixtures, we resolved to a different approach (see Experimental part for further details). 

The results point out that all investigated mixtures present a freezing point below 260 K (see Table 1), proving an unconventional behaviour. This allows us to unambiguously classify the analysed solvent as a low-melting system. However, a more detailed study is required to clarify if the 6:1 ratio effectively corresponds to the exact eutectic mixture or if a lower freezing point can be obtained by varying the hydrogen bond donor (HBD): hydrogen bond acceptor (HBA) ratio. Therefore, hereafter we will refer to them as DES-like systems, also according to what was reported by Abbott and co-workers for similar mixtures [45]. To shed more light on this aspect but limiting the analyses to the case of Gly/KBr systems, we prepared two other mixtures having a lower (Gly/KBr 4:1) or higher (Gly/KBr 10:1) amount of HBD, leading to a variation of the magnitude of HBD-HBA interaction. In both the cases, compared to Gly/KBr 6:1 (T_m_ = 225 K), higher melting temperature values (i.e., 238 K and 247 K, respectively) were measured. Therefore, in the following, we focused our analyses on systems having an HBD/HBA ratio equal to 6.

Comparing mixtures in which the halogen atom of the HBD is changed, the mixture containing the Br^−^ demonstrated the lowest melting temperature, whereas Cl^−^ and I^−^-based systems melt at higher temperatures (i.e., 239 K and 251 K, respectively). This evidence proves that the melting temperature of the mixtures cannot be directly correlated with the electronegativity of the halogen (Cl^−^ > Br^−^ > I^−^), proving the significance of the interplay between the different interactions and thus allowing the formation of a stable DES (-like) system.

Once the nature of the proposed solvent had been elucidated, a model reaction was preliminarily studied (Table 2), consisting in the reaction, at room temperature, of 4-nitrobenzenediazonium tetrafluoroborate (**1a**) in both mixtures of glycerol/KBr and conventional organic solvents. The choice of privileging the bromination reactions arises from the reason that, from the synthetic point of view, bromides, compared to chlorides, can be more useful. For example, in the cross-coupling reactions, C–Cl bonds are often too inert, and bromides (or iodides) leaving groups are required for acceptable yields [46].

Firstly, it must be stressed that, at room temperature, when adding salt **1a** (2 mmol) to a suspension of KBr in glycerol, target 1-bromo-4-nitrobenzene (**2a**) was not formed (Table 2; entry 1), not even in strong excess of KBr (Table 2; entry 2). Furthermore, the same negative result was obtained with a KBr suspension in common organic solvents (Table 2; entries 3–5). This evidence shows that a close interplay between glycerol and salt is required to activate the target reaction. Thus, homogeneous mixtures of glycerol and KBr (in various ratios, 10:1; 6:1 and 4:1) were tested. When dissolving **1a** (2 mmol) in 2.5 mL of this solution (Table 1; entries 6–8), only traces of **2a** were detected. We obtained the target **2a** (Table 1; entries 9–11), albeit still in unsatisfactory yields, using 3.5 mL of solvent systems. Consequently, we decided to double their amount (5 mL). The reaction time was 4 h; in these conditions, to our delight, a dramatic increase in yield occurred (72%. Table 1; entries 12,13). The by-product nitrobenzene (**3a**) was easily removed under vacuum. Alternatively, it could be recovered (20% yield) by chromatographing the crude residue on a short column (see Experimental). It must be stressed that the increase in the amount of solvent or concentration of KBr (Table 2, entries 14,15) does not lead to higher yields, further confirming the likely necessity of the instauration of intermolecular interactions.

After this optimization and choosing glycerol/KBr 6:1 (method A) as privileged solvent systems, the scope of this reaction was extended to a wider library of arenediazonium salts **1** containing both electron-donating and electron-withdrawing groups. The results are summarized in Figure 1.

Good yields of target aryl bromides **2** were achieved in the presence of electron-withdrawing groups. On the other hand, a decreasing of the yields was observed in the presence of electron-donating groups. In any case, longer reaction times or a greater amount of Glycerol/KBr 6:1 did not lead to sizeable improvements. It must be stressed that all aryl bromides **2** were obtained in adequate purity, making further chromatographic purification usually unnecessary since the by-products arenes **3** were usually removed under vacuum. However, from a GC-MS analysis of the crude residues, it is possible, albeit in a completely qualitative way, to have indications on the amount of **3** that were formed. The data are shown in Table 3. 

In light of the data described above, a second DES-like mixture formed by glycerol and tetrabutylammonium bromide (TBAB) in ratio 6:1 was used as an alternative solvent to glycerol/KBr (method B). As can be seen from Figure 1, the results achieved with method A or, alternatively, with method B, are almost identical. At the end of the reactions, the solvent system glycerol/KBr was easily recovered (for details see Experimental) and was used in four consecutive runs (Table 4), without observing a decrease in the yield of **2a**. In fact, potassium tetrafluoroborate, formed in the reaction as a by- product, remained trapped in the solvent system upon evaporation of the aqueous layer of the extraction, without interfering in the following runs.

In order to further expand the scope of the reaction, chloro and iododediazotation reactions were also studied. Regarding chlorodediazotation, in order to find optimal reaction conditions, a model reaction was preliminarily studied with 2,4-nitrobenzenediazonium tetrafluoroborate (**2i**) reacting at room temperature in various solvents (Table 5). 

The optimal reaction conditions (Table 5; entry 10) were found by reacting salt **1i** (2 mmol) with 10 mL of solvent system Glycerol/KCl 6:1 for 4 h. In fact, target 1-chloro-2,4-dinitrobenzene (**4i**) was obtained in a quantitative yield. After this optimization, the scope was extended to arenediazonium salts **1,** containing both electron-donating and electron-withdrawing groups. The results are reported in Figure 2. In the best conditions, salt **1j** furnished **2j** in excellent yield. It is necessary to point out that, for the success of chlorodediazotation, it is essential to have strong electron-withdrawing groups linked to the aromatic ring. In the case of the salt **1a** with a single electron-withdrawing group, a low amount of **4a** was obtained and only with an excess of Gly/KCl 6:1 (20 mL) and heating to 35 °C for 4 h. In the presence of electron-donating groups (salts **1e** and **1f**), the reaction did not occur. Similar results were obtained using a solvent system formed by glycerol and tetrabutylammonium chloride (6:1 ratio).

Finally, although the iododediazotization reaction is normally carried out in the absence of Cu,11,12, Figure 3 reports the preparation of some iododerivatives **5** reacting some arenediazonium salts **1** in Gly/KI solvent system. The yields are always virtually quantitative, independently on the nature of the substituent. 

As mentioned above, the Sandmeyer reaction has been rediscovered and “modernized” in recent years. In particular, new methodologies that allow the target products to be obtained in the absence of copper catalysts have been proposed [37,38,39,40,41]. As a result, in the light of the above experiments, we have proposed here a sustainable version of the Sandmeyer halodediazotation reaction. In fact, in our conditions, we were able to achieve important benefits, including the reaction’s use at room temperature in mild conditions and without any metal catalyst and a recoverable and reusable sustainable reaction medium, which plays the role not only of solvent but also of reagent. Moreover, the target halides are usually obtained in adequate purity, making further chromatographic purification unnecessary, and the only solid waste product is potassium tetrafluoroborate, which is soluble in water.

Beside these aspects, we found it interesting to study this reaction from a mechanistic point of view. The mechanisms for the reactions in glycerol with fluoride [10], chloride, bromide, iodide, and pure glycerol were calculated for the electron-poor **1a** and for the electron-rich **1f**
*para*-substituted benzenediazonium tetrafluoroborates. The preliminary calculations indicated that the dissociations of **1a** and **1f** to BF_4_^−^ and to 4-nitrobenzenediazonium (**1a’**) and 4-methoxybenzenediazonium (**1f’**), respectively, are thermodynamically favoured by 0.5 and 2.2 kcal mol^−1^ in terms of free energies, as expected by the ion solvation properties of glycerol. The tests for the reaction of **1a’** with fluoride suggested that the appropriate model for the computational study requires the presence of some explicit glycerol molecules. 

In the textbooks [47,48,49], the nucleophilic substitution in arenediazonium compounds is supposed to take place by the S_N_1 mechanism where the key intermediate is the phenyl cation (i.e., phenylium). However, this cation is so reactive that it can recapture the nitrogen generated in the decomposition (N_2_-scrambling) [50]. 

Additionally, attempts to observe the formation of phenyl cations by ionization of aryl triflates have only succeeded when especially stabilizing groups, such as trimethylsilyl groups, are present at the 2- and 6-positions of the aromatic ring [51]. Moreover, the experiments that prove the existence of the phenylium have all been performed in an ionizing but almost non-nucleophilic solvent such as 2,2,2-trifluoroethanol. The transition structure (TS) that we optimized for the N_2_-scrabling in 4-nitrobenzenediazonium in glycerol is located 30.6 kcal mol^−1^ above **1a’**. However, in all the attempts made to optimize this TS in the presence of explicit glycerol molecules, we found that the C-N bond breaking was immediately followed by the solvent capture (solvolysis). Therefore, the free 4-phenylium is not generated, but, after a proton transfer, the final products are the mixed ether **6a** and the protonated glycerol (Figure 2).

Thus, the solvolysis takes place through a concerted TS corresponding to a *front*-S_N_2 as illustrated in Figure 4 (up). Indeed, such a mechanism was already proposed in the literature [52,53], and it is operative also for the halogenations (Figure 3; an example of the TS is shown in Figure 4 (down), and the pictures of all structures are reported in the Appendix A). All halogenations as well as the solvolysis are irreversible, being strongly exoergic by 55–60 kcal mol^−1^ for **1a’** and by 40–56 kcal mol^−1^ for **1f’** in terms of free energies. From the activation free energies, we calculated the reaction rate constants **k** (Table 6) by means of the Eyring equation [54]. For the halogenations in Gly/potassium halide 6:1 DES-like systems, we calculated a concentration for the halogenides of 22.8 M, and from this value as well as from **k** we estimated the lifetimes **k,** which are reported in Table 5. Although underestimated because of the difficulty in modelling the solvent effects, we observed that the values of **τ** for the halogenations were in qualitative agreement with the experimental findings; the reactions of **1a’** are sensibly faster than those of **1f’**. However, in the case of fluorination, we should remember that in the Gly/KF DES (6:1 ratio) the reductions are the fastest processes [10]. For the other halogenations, because of the limitation in the calculation accuracy, the lifetimes are too close to perform any reliable comparison among the halogens. For iodination we could not exclude radical mechanism that we did not analyse, being out of the scope of the present paper. The solvolyses, which are treated as monomolecular processes as a result of the nucleophile being the solvent, show low **k** and long lifetimes, in agreement with the fact that mixed ethers **6a** and **6f** are never found in the experiments because the other reactions always prevail.

We also stress that the observation of a secondary deuterium kinetic isotope effect (KIE) of 1.49 in the dediazotation of benzenediazonium [55], interpreted as evidence of the formation of the phenylium, is also compatible with the present *front*-S_N_2 mechanism. In fact, the calculated KIE values for the solvolysis of the 2,6-dideuterated **1a’** and **1f’** are, respectively, 1.40 and 1.43. This is because the electronic and molecular structures of the aryl moieties in the transition structures are similar to that of the arylium. For the solvolysis of **1a’**, for example, the natural charges on C^1^ (0.034 in the 4-nitrobenzenediazonium) are 0.551 in the TS and 0.649 in the 4-nitrophenilium. The C^2^C^1^C^6^ angles (which is 125.4 degree in the 4-nitrobenzenediazonium) are 140.0 degrees in the transition structure and 149.3 in the 4-nitrophenilium. 

The reduction of the arenediazonium in DES (see [10]) involves the generation of a phenyl radical Ar• that abstracts a hydrogen from the solvent by a Hydrogen Atom Transfer (HAT). As an alternative, the radical can bind a halide ion X^−^, starting a radical chain that yields to the aryl halide Ar-X (Figure 4):

The first step of the radical halogenation is in competition with the HAT that yields to the reduction products. The calculated pseudo-first order rate constants for the binding of the 4-nitrophenyl radical with fluoride, chloride, bromide, and iodide are, respectively, 10^2^, 10^5^, 10^6^, and 10^8^ s^−1^, to be compared with 10^9^ s^−1^ for the HAT [10]. Clearly, the halogenation by a radical mechanism is not feasible for the three lighter halogens, but it is a possible alternative for the iodination where the radical chain can be initiated by the reduction of the arenediazonium by iodide, as proposed in the literature [11,12]. For the electron-rich 4-methoxybenzenediazonium, the chlorination and the bromination (which are the main interest of this work) by the radical-chain pathway appears to be even less feasible. The reaction of the 4-methoxyphenyl radical with chloride and bromide are strongly endoergic (in term of free energies, respectively 30.3 and 29.5 kcal mol^−1^ compared to −11.9 and −10.4 kcal mol^−1^ for the reaction with the 4-nitrophenyl radical). Therefore, the radical chain for these halogenations cannot proceed. 

Finally, we studied the mechanism for the direct reaction of **1a’** with BF_4_^−^ (an intramolecular reaction from the point of view of **1a**). The rate constant for this exoergic reaction (Δ*G* = −34.4 kcal mol^−1^) is 2.0 × 10^−7^ s^−1^, which corresponds to a lifetime of 57 days. This value is in agreement with the experiments where this reaction was performed for 4-*t*butybenzenediazonium tetrafluoroborate in CH_2_Cl_2_, yielding the corresponding fluoride with a yield of 50% ca. after 10 days at room temperature [56].

## 3. Materials and Methods

**General****.** All the reactions were carried out in open air glassware. Analytical-grade reagents and solvents (purchased from Merck or Alfa Aesar) were used, and reactions were monitored by GC, GC-MS, and TLC. Column chromatography and TLC were performed on Merck silica gel 60 (70–230 mesh ASTM) and GF 254, respectively. Petroleum ether refers to the fraction boiling in the range 40–70 °C. Room temperature is 22 °C. Mass spectra were recorded on an HP 5989B mass selective detector connected to an HP 5890 GC with a methyl silicone capillary column. GC analyses were performed on a Perkin Elmer AutoSystem XL GC with a methyl silicone capillary column. ^1^H NMR and ^13^C NMR spectra were recorded on a Jeol ECZR spectrometer at 600 and 150 MHz, respectively. Arenediazonium tetrafluoroborates were prepared as reported in the literature [57]. Structures and purity of bromoarenes **2**, chloroarenes **4,** and iodoarenes **5** were confirmed by their spectral (NMR, MS) and physical data, substantially identical to those reported in the literature. Their NMR spectra are reported in the Appendix A. Differential Scanning Calorimetry (DSC) experiments were performed on Q200 DSC TA instruments with a ramp of 5 or 20 °C/min under N_2_ atmosphere. After a stabilization at −85 °C, the sample was heated up to 80 °C and then cooled down to -80 °C, and the cycle was repeated two times. As a result of the DSC being unsuitable for the determination of its melting point in glycerol-based mixtures, we resolved to an unconventional approach: we inserted a thermometer into a vial containing the mixture, and then the vial was moved to a freezer (at −79 °C) for 15 min. After this time, the vial was removed from the fridge and was allowed to warm up to room temperature. The T_m_ value was selected as the temperature signed by thermometer when the latter was extracted (with no solid residues on it) from the vial. 

**Preparation of solvent systems: typical procedure for glycerol/KBr 6:1.** KBr (11.9 g, 0.1 mol) was added at room temperature to glycerol (55.2 g, 0.6 mol). The suspension was stirred at 80 °C for about 2 h. It was cooled to room temperature, and a clear solution was obtained, which was used without any further purification. 

**Bromodediazotation of 4-nitrobenzenediazonium tetrafluoroborate (1a): typical procedure. Preparation of 1-bromo-4-nitrobenzene (2a).** 4-Nitrobenzenediazonium tetrafluoroborate (**1a**, 0.44 g, 2 mmol) was added at room temperature to glycerol/KBr 6:1 (5 mL). The mixture was stirred at room temperature for 4 h, and the completion of the reaction was confirmed by the absence of azo coupling with 2-naphthol. Then, the reaction mixture was poured into Et_2_O/H_2_O (10 mL, 1:1). The aqueous layer was separated and extracted with Et_2_O (5 mL). The combined organic extracts were washed with H_2_O (5 mL), dried with Na_2_SO_4_, and evaporated under reduced pressure. GC-MS analyses of the crude residue showed a mixture of **2a**, as the major product, MS (EI, 70 eV): m/z (%) = 201 (100) [M]^+^, 203 (100) [M +2]^+^ and nitrobenzene **3a** MS (EI, 70 eV): m/z (%) = 123 (100) [M]^+^. Further evaporation at reduced pressure allowed **3a** to be completely removed and pure **2a** to be obtained (GC, GC-MS, TLC and NMR; 290 mg, 72%). Alternatively, in order to quantify **3a,** the crude residue was chromatographed on a short column (silica gel; eluent: PE). The first eluted product was **3a** (50 mg, 20%). The second one was **2a** (284 mg, 70%).

**Recovery and reuse of solvent system glycerol/KBr 6:1** The aqueous layers (about 15 mL) were collected and gathered. In order to remove solid residues, they were filtered on a funnel. H_2_O was evaporated under reduced pressure. The recovered glycerol/KBr (4.9 mL), which showed NMR and IR spectra virtually identical to the initial one, was reused in four consecutive reactions. The average yield of **2a** was 69%, and the recovered solvent system (at the end of fifth run) continued to show NMR and IR spectra almost identical to the initial one. 


**Computational method.**


The details and the references related to the computational method are all reported in the Appendix A.

## 4. Conclusions

We have proposed here a synthetic and computational study of halodediazotation reactions (a Sandmeyer’s class of reactions) carried out in deep eutectic solvent-like mixtures. The computational results are coherent with the experimental findings; the reactions of the control electron-poor **1a** are faster than that of the control electron-rich **1f**, and the halogenations are always much faster than the solvolysis that would yield the mixed ethers **6a** and **6f**. Both halogenations and the solvolysis take place through a mechanism similar to that of front-S_N_2. Diazonium salts are resourceful building blocks in organic synthesis, and the last few years have seen a dramatic resurgence of their chemistry, as new functionalities have been introduced on their aromatic ring. The marriage between diazonium salts and deep eutectic solvents, which have also been defined as “the solvents of the century” [5], can become not only a valid tool to make their chemistry more sustainable but can also allow the exploration of new types of hitherto unknown reactivity.

## Data Availability

The data reported in this article can be obtained from the authors upon reasonable request.

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
