# Peer review of "Copper-Free Halodediazoniation of Arenediazonium Tetrafluoroborates in Deep Eutectic Solvents-like Mixtures"

_molecules, 2022, doi:10.3390/molecules27061909_

Round 1

Reviewer 1 Report

The manuscript (molecules-1629013) by Dughera and co-workers describes Copper-free halodediazoniation of arenediazonium tetra-fluoroborates in deep eutectic solvents-like mixtures. The solvents are fully characterized and a computational study is presented aiming to understand the reaction mechanism. Overall, before publication with minor revisions.

  • In page 12,line 386. "obtained (GC, GC-MS, TLC and NMR; 290 mg, 72%),however, in page 12,line 388. " The second one was 2a (284 mg, 705) ". The yield shoud be checked in text .
  • In SI, page 94, (from MeOH. Lit.9 46–48°C). What is meaning of “from MeOH”
  • In SI, page 112. The NMR spectra of 4i is not clear, please replace new 1HNMR and 13CNMR spectra. In page 113. The 1H NMR spectra of chemical structures of 4i should be chemical structures of 4j. Please check it.
  • In page 114. The 1H NMR, 13C NMR, 19F NMR spectra of 4 m should be added.

Author Response

Referee 1

In page 12, line 386. "obtained (GC, GC-MS, TLC and NMR; 290 mg, 72%),however, in page 12,line 388. " The second one was 2a (284 mg, 705) ". The yield shoud be checked in text

First of all, there is a typo. In fact 705 is clearly wrong and should be read as 70%. Going into detail, there are two different reactions: The first one was carried out to isolate the target bromide only (obtained with a 72% yield). In the second one, however, we also wanted to isolate the reduction product. For this reason, the yields are not identical, but still very similar.

In SI, page 94, (from MeOH. Lit.46–48°C). What is meaning of “from MeOH

We mean: crystallized from methanol. In the SI, we have always added the word “crystallized”.

In SI, page 112. The NMR spectra of 4i is not clear, please replace new 1HNMR and 13CNMR spectra. In page 113.

In order to make the spectrum clearer, we have added in SI a highlight by expanding the area where all the various signals are present.

The 1H NMR spectra of chemical structures of 4i should be chemical structures of 4j. Please check it

By mistake the 4i formula was insert on the spectrum, instead of that of 4j. Actually, the spectrum is correct and it is precisely that of 4j. We simply corrected the formula.

In page 114. The 1H NMR, 13C NMR, 19F NMR spectra of 4 m should be added.

We added them.

Reviewer 2 Report

The manuscript of the article entitled "Copper-free halodediazoniation of Arenediazonium Tetra-fluoroborates in Deep Eutectic solvents-like mixtures." by Giovanni Ghigo, Matteo Bonomo, Achille Antenucci, Chiara Reviglio and Stefano Dugher is interesting for readers of Molecules and as it extends the current level of knowledge.  I have following comments and requirements:  The authors present the 100% yields (Cart 1, Chart 2), which I don't think is appropriate. A much better expression is using the term quantitative yield. For a detailed discussion of the presentation of synthesis yields refer account by Prof. Hudlicky and Wernerova (Synlett 2010, 18, 2701). It is not clear whether the authors isolated intermediate 6a (Scheme 2). For express the proposed reaction mechanism, I recommend displaying the reaction coordinate with indicating the structures and energies for the individual transition states and the intermediates. Did the authors proposed the possibility of coordination of the arene cation to two electron rich neighboring glycerol hydroxygroups? The resulting associate would form a relatively stable five-membered cycle. The present manuscript should be acceptable after minor revision for publishing in the Molecules.

Author Response

Referee 2

The authors present the 100% yields (Cart 1, Chart 2), which I don't think is appropriate. A much better expression is using the term quantitative yield.

We did it

For a detailed discussion of the presentation of synthesis yields refer account by Prof. Hudlicky and Wernerova (Synlett 2010, 18, 2701).

The yields of the adducts in Charts 1,2,3, as reported in the notes, refers to pure and isolated products. As far as table 3 is concerned, our aim, above all, was to show that a certain amount of reduction product is always formed alongside the target halogenation product.

Reduction products are often volatile, so it is impossible to quantify them. We therefore tried to estimate their quantity through GC analysis, without expecting to give their yield with absolute certainty.

It is not clear whether the authors isolated intermediate 6a (Scheme 2).

The mixed ethers like 6a and 6f (and all others) are never found in the reaction mixture coherently with the calculated rate constants. Actually, we already made explicit it in the text (page 10, lines 324-328) and in the conclusions (page 13, lines 432-434).

For express the proposed reaction mechanism, I recommend displaying the reaction coordinate with indicating the structures and energies for the individual transition states and the intermediates. 

Both solvation and halogenations are one-step reactions, so there are no intermediates.  Therefore, reaction energy profiles would not be useful to catch the nature of the mechanisms.  Anyway, to clarify this aspect, Scheme 2 has been revised and a new Scheme, 3, has been added.

We believe that for a semi-quantitative comparison between the solvation and the halogenations and between the electron-poor 1a and the electron-rich 1f, the calculated rate constants are more useful. 

Did the authors proposed the possibility of coordination of the arene cation to two electron rich neighboring glycerol hydroxygroups? The resulting associate would form a relatively stable five-membered cycle

Indeed, all reactions are studied with supramolecular models that all  include some explicit glycerol molecules. This is stated in the manuscript (page 9, lines 276 and 288). More details are reported in the SI.